health and disease and epidemiology/computer modelling and simulation

infectious disease, mumps outbreak, college campus, stochastic susceptible exposed infectious recovered model, public health intervention, Harvard University

**Author for correspondence:**
Andrés Colubri
e-mail: andres.colubri@umassmed.edu

# Containing the spread of mumps on college campuses

Mirai Shah[1], Gabrielle Ferra[2], Susan Fitzgerald[3], Paul J. Barreira[4], Pardis C. Sabeti[5,6,7,8] and Andrés Colubri[5,6,7,9]

[1]Harvard College, Cambridge, MA, USA
[2]Brown University, Providence, RI, USA
[3]Harvard University Health Services, Cambridge, MA, USA
[4]Harvard Medical School, Boston, MA, USA
[5]Department of Organismic and Evolutionary Biology, Harvard University, Cambridge, MA, USA
[6]Broad Institute of MIT and Harvard, Cambridge, MA, USA
[7]Howard Hughes Medical Institute, Chevy Chase, MD, USA
[8]Harvard School of Public Health, Boston, MA, USA
[9]Program in Bioinformatics and Integrative Biology, University of Massachusetts Medical School, Worcester, MA, USA

AC, 0000-0001-5559-9661

College campuses are vulnerable to infectious disease outbreaks, and there is an urgent need to develop better strategies to mitigate their size and duration, particularly as educational institutions around the world adapt to in-person instruction during the COVID-19 pandemic. Towards addressing this need, we applied a stochastic compartmental model to quantify the impact of university-level responses to contain a mumps outbreak at Harvard University in 2016. We used our model to determine which containment interventions were most effective and study alternative scenarios without and with earlier interventions. This model allows for stochastic variation in small populations, missing or unobserved case data and changes in disease transmission rates post-intervention. The results suggest that control measures implemented by the University's Health Services, including rapid isolation of suspected cases, were very effective at containing the outbreak. Without those measures, the outbreak could have been four times larger. More generally, we conclude that universities should apply (i) diagnostic protocols that address false negatives from molecular tests and (ii) strict quarantine policies to contain the spread of easily transmissible infectious diseases such as mumps among their students. This modelling approach could be applied to data from other outbreaks in college campuses and similar small population settings.

# 1 Introduction

College campuses provide ideal breeding grounds for infectious disease. Students live in close quarters, pack into lecture halls, share food and drinks in the dining areas and engage in intimate contact. Outbreaks in these settings can spread very quickly. Indeed, a meningitis outbreak took place at Princeton University in March 2014, eventually claiming the life of one student. The Centers for Disease Control and Prevention reported the attack rate of the disease on Princeton's campus to be 134 per 100 000 students—400 times greater than the national average [1]. Recent spread of COVID-19 in educational settings [2] forced school closures around the world [3] and motivated the design and implementation of plans for safe reopening [4,5].

A recent string of outbreaks on college campuses involves mumps, once a common childhood disease caused by the mumps virus [6]. After introduction of the measles-mumps-rubella (MMR) vaccine in 1977 and the two-dose MMR vaccination programme in 1989, the number of mumps cases in the United States plummeted by 2005. But, despite a vaccinated population, there has been a recent resurgence of mumps, with a steep jump from 229 cases in 2012 to 5833 cases in 2016 [7]. Although a typically mild disease in children, up to 10% of mumps infections acquired after puberty can cause severe complications, including orchitis, meningitis and deafness. Furthermore, most recent mumps cases have occurred in young adults who had received the recommended two MMR doses. This suggests that vaccine-derived immunity wanes over time, unlike natural immunity—protection acquired from contracting the disease—which is either permanent or wanes more slowly. Lewnard and Grad [8] estimate that 33.8% of young adults (ages 20 to 24) were susceptible to mumps in 1990, in contrast to the 52.8% susceptible in 2006, as vaccinations have replaced contraction as the source of immunity. The proportion susceptible among college-age individuals has stabilized in the past ten years since most individuals in this age group have received two MMR doses, and the waning immunity is thought to occur after the second dose. The temporary immunity from vaccines strengthens the argument for strict containment as a critical line of defense amidst an outbreak. As illustrated by the COVID-19 pandemic, the challenges associated with wide and equitable distribution of vaccines [9], substantial asymptomatic and pre-symptomatic transmission of the disease [10] and the possibility of new viral strains with higher transmissibility [11] provide support for such approaches.

The spread of mumps at Harvard University in 2016, and extensive public health measures and documentation, presents an opportunity to closely examine an outbreak on a college campus. Between 1 January 2016, and 31 August 2016, 210 confirmed mumps cases were identified in the Greater Boston area, with most detected at Harvard University. Mumps is a highly contagious disease with the potential to travel quickly and pervasively on a crowded college campus. Some of the most notable mumps outbreaks on college campuses occurred in Iowa [12], Indiana [13] and Ohio [14]. But, whereas mumps spread rapidly at Ohio State University in 2014 and the University of Iowa in 2006 and 2016, Harvard employed a number of interventions that may have helped mitigate spread of the disease and contain it over just a few months [15]. The possibility of distinct viral strains resulting in different outbreak dynamics between schools is unlikely, as it was shown by application of genetic epidemiology methods [16,17] that all mumps outbreaks in the United States since at least 2006 have been likely caused by the same lineage, mumps virus genotype G.

The successful containment at Harvard motivates us to explore varied intervention strategies, given the relative costs of prevention. Even if the use of a booster MMR vaccination is proven theoretically to reduce infection and thus potentially prevent outbreaks [8,12], it is unlikely that universities with limited resources will proactively invest in a third dose. A rough cost analysis conducted by Harvard University Health Services (HUHS) showed that, while the total mumps care expenses for Harvard was approximately $75 000, the cost of providing a third MMR dose to every member of the Harvard community (at $83 per dose) was $1.7 million [18]. Therefore, at least in the short term, a third MMR dose cannot be the only answer to handling mumps outbreaks; we must consider more immediate solutions and interventions.

With the goal of understanding the effectiveness of interventions aimed at containing mumps outbreaks on a college campus, we constructed an epidemiological model to simulate the dynamics of mumps on such a population and quantify the impact of various interventions. We surveyed the literature, but we were not able to find epidemiological modelling studies of mumps in such settings. This modelling can be challenging because the number of susceptible students in a college is small compared with nation-wide or even regional studies, parameters characterizing the interventions are not known, and data are partially observed. Furthermore, deterministic compartmental models are only appropriate when the populations of the compartments are sufficiently large [19]. We adopted a

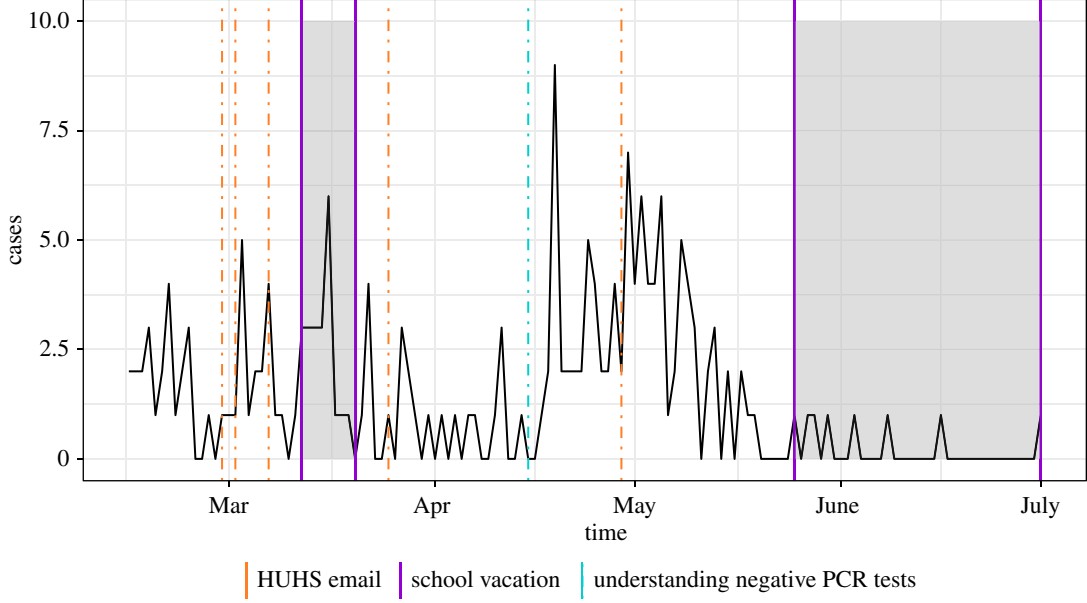

**Figure 1.** The daily number of new mumps cases at Harvard and the timeline of school vacations and control interventions employed by HUHS between February and June 2016. HUHS sent multiple emails over the course of the outbreak, raising awareness about the spread of mumps. Additionally, in mid-April, HUHS began more carefully diagnosing mumps, rather than automatically ruling out those with negative PCR tests. The isolation policy is not shown because it occurred continuously throughout the entire outbreak.

stochastic susceptible exposed infectious recovered (SEIR) model presented to address these issues. We developed this model within the framework of partially observed Markov processes (POMPs), which has been applied to introduce structural stochasticity into epidemic models [20]. The stochastic nature of the model allows for variability from elements that are not modelled explicitly, such as class schedules and campus layouts. This model also allows us to easily quantify time-varying interventions after fitting the parameters to the observed data, by running simulations under alternative scenarios.

We fitted model parameters on case data for Harvard's 2016 mumps outbreak provided by the Massachusetts Department of Public Health (MDPH). In applying our model, we were able to quantify the effect of the interventions employed by HUHS—more aggressive diagnosis protocols where clinical symptoms alone were enough to result in quarantine, and strict isolation of suspected cases—on reducing the size and duration of the outbreak. Given the lack of prior modelling studies of mumps outbreaks on college campuses, our work represents a novel application of POMP models to such settings. The conclusions from this paper could be useful to guide future responses to outbreaks of mumps and other infectious diseases in higher education institutions. Without effective measures in place, diseases like mumps, meningitis and now COVID-19 have the potential to spread in these environments with ease and lead to serious health complications. Simple interventions that ensure most cases are detected, treated and separated from susceptible individuals make a significant difference.

# 2. Material and methods

## 2.1. Outbreak data

The mumps outbreak at Harvard began in mid-February 2016, when six students reported onset of parotitis to HUHS. For the next three months, the number of cases continued to rise, until finally plateauing in late May and early June. There were two waves of the outbreak—one occurring in the month of March and a larger one occurring in mid-April—totaling 189 confirmed and probable cases (figure 1). Confirmed cases are those with a positive polymerase chain reaction (PCR) test for mumps

virus. Probable cases are those who either tested positive for the anti-mumps IgM antibody or had an epidemiologic linkage to another probable or confirmed case (Barreira P, Fitzgerald S. 2018, personal communication) [21]. Suspected cases were identified by presentation of the following clinical symptoms, per MDPH guidelines [22]: low-grade fever, swelling of one or more salivary glands, usually the parotid gland and/or orchitis, and a prodrome consisting of myalgias, loss of appetite, malaise and/or headache. The majority of these cases received the recommended two doses of MMR prior to contraction of mumps infection [23].

We use data provided by MDPH, which documented every mumps case between 2015 and 2017 at schools across Massachusetts [24]. These data include demographics of the patient (gender, age, county, and institution), symptoms and vaccination status, date they reported their symptoms and the date of symptom onset and lag time between the date of symptom onset and admission to a medical clinic. In our calculations, we use data from 14 February 2016, the date of the first reported case that is considered part of the outbreak (day 0), until 1 September 2016 (day 200).

## 2.2. Containment interventions

Harvard University employed three main interventions to contain the outbreak: (i) awareness raising, (ii) case identification and testing and (iii) isolation of infectious individuals. First, between February and May 2016, HUHS sent six different emails to Harvard students, employees and colleagues with information on the gravity of the outbreak, recommendations on how to prevent transmission and instructions on how to identify mumps. This raised awareness throughout the campus. Particularly at the peak of the outbreak, roommates, resident deans and athletic coaches all played essential roles in reporting potential cases of mumps, so that few cases likely went undetected and untreated by HUHS (P. Barreira, S. Fitzgerald 2018, personal communication) [21]. Second, Harvard acted vigorously to treat and isolate anyone suspected of mumps throughout the outbreak. Initially, due to the disease's non-specific symptoms and less extreme manifestation in vaccinated people, HUHS used positive mumps PCR tests as a necessary ground for diagnosis. Later, on recommendation from the MDPH, HUHS stopped automatically ruling out those with negative PCR results, given that false negatives were quite frequent in vaccinated individuals and that some individuals reported their infection to the clinic belatedly. Detection of the mumps virus by molecular testing is challenging because of its transient replication [25]. In outbreaks among two-dose vaccine recipients, mumps virus was only detected in samples from approximately 30–35% of case patients if the samples were collected within the first 3 days following onset of parotitis [26]. Anyone who entered HUHS displaying clinical symptoms of mumps was now deemed infected and infectious. This change in the diagnosis protocol took place on 15 April 2016, day 61 of the outbreak [21]. Third and perhaps most notably, Harvard isolated most confirmed or probable cases of mumps. While many universities simply suggest self-isolation in one's room or dormitory (which could expose roommates and friends to the disease), Harvard removed anyone with clinical symptoms of mumps from the population. Of the 230 total cases at Harvard between February 2016 and November 2017, 96 were isolated in alternate housing on campus, while 110 were isolated off-site. Although a person remains infectious with mumps for 5 days, Harvard isolated patients for 6 days for additional measure (P. Barreira, S. Fitzgerald 2018, personal communication).

Harvard also used a variety of smaller interventions to contain the disease. For instance, water fountains with a weak upward flow were repaired in late March when it became apparent that students were directly touching the fountain with their water bottles or mouths [21]. In this study, we only considered the three larger scale interventions. Figure 1 shows a timeline of the interventions as well as periods when the population was fluctuating (such as during spring and summer break). Around two weeks after HUHS updated its criteria for diagnosis in mid-April, there was a steep decline in the number of new cases. Regarding the email awareness campaign, Harvard implemented this campaign via emails regularly sent out by HUHS recommending personal hygiene and testing in case of symptoms compatible with mumps. Anecdotal evidence (i.e. conversations with students) and, most importantly, the fact that HUHS emails were sent regularly throughout the outbreak made us conclude that emails were not particularly effective to limit spread of the outbreak but could have contributed to the general awareness and high reporting rate we noted earlier. Thus, we evaluate the effects of the two main interventions (stringent diagnosis protocol and strict isolation of suspected cases) further in the modelling section of this paper. We should note that interventions were possible thanks to the ample resources that Harvard has at its disposal, which may not be available at other

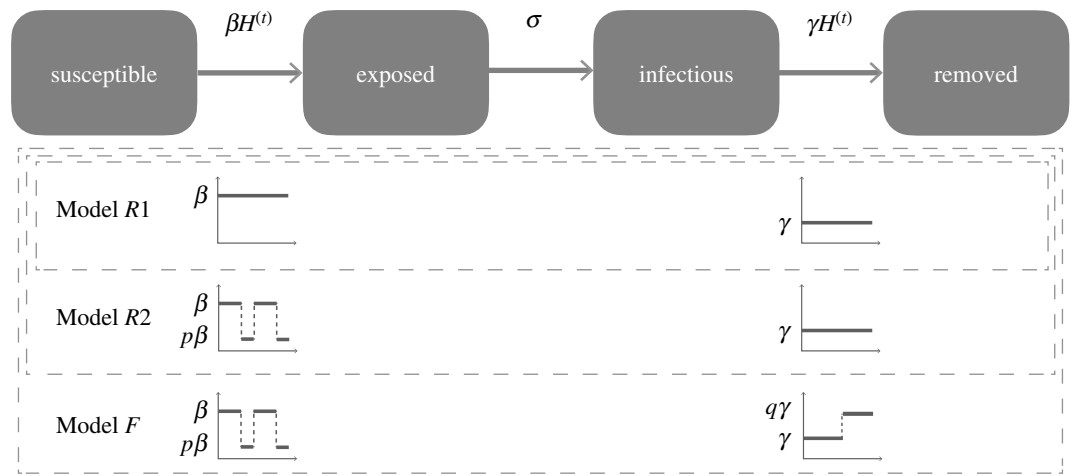

**Figure 2.** Schematic of the SEIR models used in our analysis of the data, showing the compartments at the top, and the three different models below. These models all share the same compartment structure but differ in their time-dependent transition and removal rates, with model $R1$ having all rates constant, model $R2$ introducing a transmission rate that decreases by a factor of $p$ during school vacation and model $F$ increasing the removal rate by a factor of $q$ due to the change of diagnosis protocol by HUHS. These models define a nested sequence, with model $F$ containing model $R2$ (when $q = 1$), and model $R2$ containing model $R1$ (when $p = 1$).

universities. Nevertheless, this situation makes Harvard an ideal testing ground for interventions that could not be deployed elsewhere, at least without solid proof of their efficacy.

## 2.3. POMP SEIR model

We adopted a parsimonious SEIR compartmental structure [27] without any additional compartments or interactions to model the mumps outbreak at Harvard University (figure 2). The residential nature of the school and the limited duration of the outbreak (which ended before the next cohort of students arrived on campus) give support to the adoption of a simple SEIR structure, as the population size was approximately constant and interactions with the non-student population were limited. We built our SEIR model within the POMPs framework [28], which allowed us to introduce stochasticity in the epidemic dynamics and handle incomplete case data, which is needed since not all mumps cases are reported, and latent mumps carriers exhibit no symptoms at all. POMP models require the specification of a process model that describes stochastic transitions between the (unobserved) states of the system (the SEIR compartments) and a measurement model where the distribution of observed data (confirmed cases) is a function of the unobserved state variables. (Here, we only provide an overview of our modelling approach, please refer to the electronic supplementary materials, for additional details.)

We defined the process model as a stochastic SEIR where the counts of new susceptible, infectious and removed cases are drawn from binomial distributions. This model has time-varying parameters to represent different assumptions concerning the timing of the interventions and changes in interaction patterns between students throughout the school year. These parameters are $\beta(t)$, the transmission rate of the outbreak, and $\gamma(t)$, the removal rate between the infectious and removed compartments (figure 2). Other parameters, such as the population size $N$ or $\sigma$, the transition rate between the exposed and infectious compartments, are known from either Harvard's enrolment and calendar data, or the biology of mumps, and therefore fixed (table 1 for a list of all fixed parameters in the model). An important assumption in our fixed parameters is the use of an effective population size $N_{eff} = N \times 0.53$. In this formula, $N$ is the total school population [30] including undergraduates and students enrolled in all graduate schools and extension programmes, because the outbreak, even though it mainly affected undergraduate students, also spread among other subgroups [16]. The 0.53 factor is the estimated fraction of mumps-susceptible individuals between 20 and 29 years of age in the United States. [8]. Even though the school population at Harvard likely included some individual above (staff) and below (undergrads) that age range, we consider this limitation in our model to be minor as most individuals were within it. We picked an over-dispersed binomial for the distribution of the number of observed cases given the number of true cases. This distribution has the following

**Table 1.** List of all parameters in the full model, including fixed parameters that are determined by mumps biology and the school's enrolment and calendar and the parameters obtained by MLE or calculated using the estimated parameters.

| symbol | description | value | 95% CI | units | source |
|---|---|---|---|---|---|
| $\tau$ | date of intervention | 61 | — | d | fixed: from records on interventions [21] |
| $t_0, t_1, t_2$ | vacation dates, 2015–2016 academic year | 26, 34, 100 | — | d | fixed: from archived academic calendar [29] |
| $\sigma^{-1}$ | duration of mumps latent period | 17 | — | d | fixed: from Lewnard and Grad [8] |
| $\gamma^{-1}$ | duration of mumps recovery period | 5 | — | d | fixed: from Lewnard and Grad [8] |
| $N_{eff}$ | effective population | 10 600 | — | — | fixed: from University enrolment records [30] and mumps susceptibility among college-aged individuals [8] |
| $\beta$ | baseline transmission rate | 1.39 | (1.02, 2.20) | $d^{-1}$ | MLE |
| $\gamma$ | baseline removal rate | 0.85 | (0.78, 0.99) | $d^{-1}$ | MLE |
| $p$ | decrease in infection due to vacation | 0.11 | (0.00, 0.47) | — | MLE |
| $q$ | increase in removal rate | 2.8 | (1.39, 6.00) | — | MLE |
| $\rho$ | proportion of infections reported | 0.97 | (0.87, 0.99) | — | MLE |
| $\psi$ | overdispersion parameter | 0.54 | (0.36, 0.72) | — | MLE |
| $R_E(t)$ | effective reproduction number | 1.63 (normal term) 0.18 (vacation) 0.58 (post intervention) | — | — | calculated as $\frac{\beta(t) S(t)}{\gamma(t) N_{eff}} \approx \frac{\beta(t)}{\gamma(t)}$ |

parameters: the reporting rate $\rho$ ($\rho < 1$), and the factor $\psi$ that models variability in small populations such a college campus.

Given this general structure, we compared three modelling hypotheses about the nature of the on-campus interactions and the interventions by HUHS: constant rates $\beta$ and $\gamma$ of transmission and removal implying no changes in interactions and no effect of interventions (denoted as reference model 1 or $R1$ for short), variable rate $\beta(t)$ representing lower interaction levels during school vacation but constant $\gamma$ (denoted as model $R2$) and variable rates $\beta(t)$ and $\gamma(t)$ reflecting relaxed interaction outside of school terms and the effect of the updated diagnosis and isolation protocol (denoted as 'full' model or $F$). More concretely, we propose a transmission rate $\beta_H(t) = p\beta\_$ if $t0 \leq t \leq t1$ or $t \geq t2$, $\beta$ otherwise. In this formula, $t0$ and $t1$ are the starting and ending dates for the spring break, $t2$ the beginning of the summer recess, $\beta$ is the baseline transmission rate during normal class term and $p$ is a number between 0 and 1 that accounts for the reduction in exposure contacts within the student population during the school vacations. We also propose a varying removal rate $\gamma_H(t) = q\gamma\_$ if $t \geq \tau$, $\gamma$ otherwise, where $q$ is a constant greater than 1 and $\tau$ is the date when HUHS diagnosis protocol was updated to consider clinical presentation of symptoms alone enough for strict isolation of suspected cases. Models $R1$, $R2$ and $F$ define a nested sequence of models in parameter space, with $H^{R1} = \mathcal{M}(\beta,\gamma,\rho,\psi) \subset H^{R2} = \mathcal{M}(\beta,\gamma, p,\rho,\psi) \subset H^F = \mathcal{M}(\beta,\gamma,p, q,\rho,\psi)$. In such a scenario, we can apply the likelihood ratio test [31] to select the best model according to maximum-likelihood estimate (MLE) yielding the parameter values that maximize the log likelihood of the observed data given to each model. Within the POMP framework, we can perform fast MLE via sequential Monte Carlo techniques [28] to find the optimal parameters that fit each model to the observed data.

## 2.4. Intervention analysis

We performed an analysis of the parameter $q$ that represents the effect of what we consider to be the defining intervention at Harvard—updated diagnosis protocol—occurring around day 61 of the outbreak. This could allow us to understand to what extent this intervention made a difference on the trajectory of the outbreak. First, we compared the scenario with the intervention versus a scenario without the intervention. Keeping all other parameters fixed, we ran two sets of simulations at the MLEs, with 200 simulations each. The first set of simulations had $q$ fixed at the value obtained from MLE, while the second set of simulations set $q$ to 1, reflecting that no interventions occurred around day 61. We then compared the cumulative number of cases over time for these two sets of simulations, generating a 95% percentile range from all the simulations in each set. Second, we used this method to determine if administering the interventions earlier could have lowered the number of cases. We let the day of the intervention take on values between 1 and 60. Subsequently, we ran simulations for each of these 60 cases, pulled the final outbreak size from the median simulation and calculated the reduction in outbreak size.

# 3. Results

## 3.1. Model selection and MLEs

The null hypotheses of the simpler models $R1$ and $R2$ compared against the full model $F$ are both rejected at the 95% level by application of the likelihood ratio test (see electronic supplementary materials, for the log likelihood values of each model). The MLEs for all the parameters in the full model are all listed in table 1, including intervention parameter $q$, baseline rate of removal $\gamma$, reporting rate $\rho$, overdispersion parameter $\psi$, and effective reproduction number $R_E$.

We approximated $R_E$ by $\beta_H(t)/\gamma_H(t)$, which holds in our case because the maximum total number of cases across all our simulated runs is small compared with $N_{eff}$ (480 versus 10 600). Since $\beta(t)/\gamma(t) \times S_{min}/N_{eff} \leq R_E(t) \leq \beta(t)/\gamma(t)$, where $S_{min}$ is the minimum number of susceptibles over all simulations (see electronic supplementary materials for details), we get, for example, $1.63 \times 0.95 = 1.55 \leq R_E(t) \leq 1.63$ for the normal school term, showing that our approximation is justified.

The corresponding tables for the reference models $R1$ and $R2$ are provided in the electronic supplementary materials (electronic supplementary material, tables S1 and S2). Since these models had to fit the observed data without the intervention that increased the removal rate, their MLE of the transmission rates is lower to compensate. Finally, we also fitted a full model with a lower $N_{eff}$ representing the situation where all international students are fully immunized against mumps,

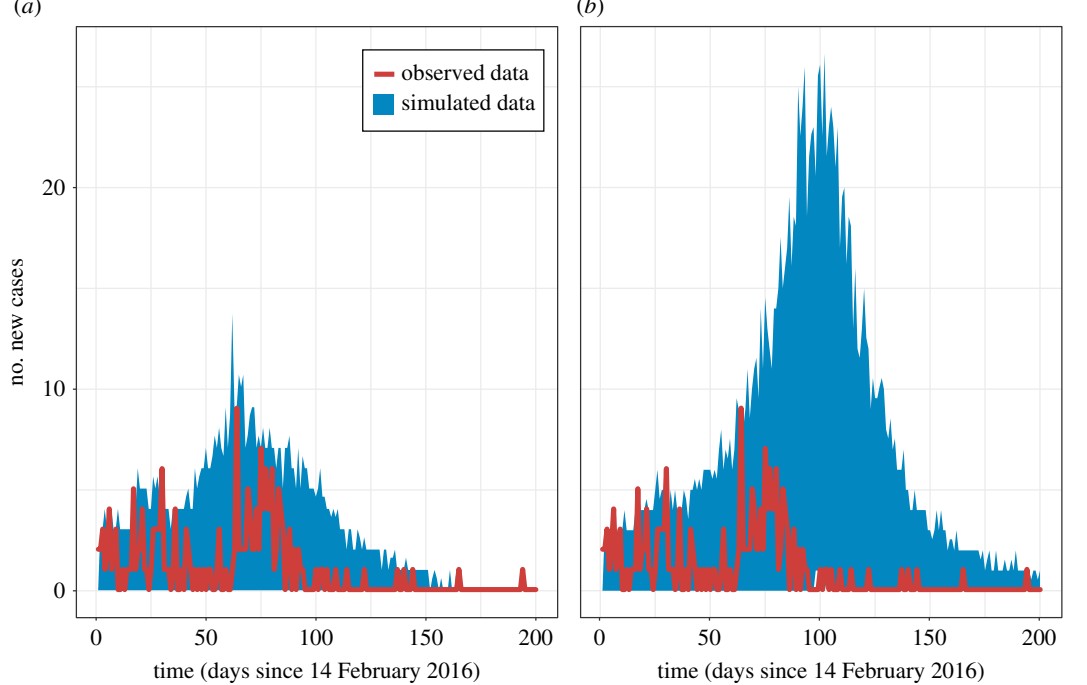

**Figure 3.** Plots showing the observed case count data (red line) and the range of simulated case count values at each time point between the bottom 5% and top 95% percentiles (blue shaded area) from 200 simulation runs using the full model (*a*) and the full model without increase of removal rate (*b*).

denoted $F^*$, and its MLEs are presented in electronic supplementary material, table S3. All the estimates fall within the 95% CIs of the original interaction model, except for the removal rate $\gamma$, which falls slightly to the left of the CI.

## 3.2. Simulation runs

We ran 200 stochastic simulations using the MLE parameters for each one of the three models (reference $R1$ and $R2$ and full model $F$) from day 0 until day 200. We also considered a fourth model derived from the full model, where we set $q$ to 1 to simulate the absence of intervention under the same parameters derived from the observed data assuming intervention. Figure 3 shows the range of values between the bottom 5% and top 95% percentiles from all 200 simulations. Electronic supplementary material, figure S1 depicts nine individual simulation trajectories for each model to illustrate the variability across the simulations.

The full model (figure 3*a*) exhibits good agreement with the data. Shortly after day 61 (the time of the intervention), the number of cases begins to decrease, reaching zero by day 150. The reference models $R1$ (electronic supplementary material, figure S2A) and $R2$ (electronic supplementary material, figure S2B) show lower case counts during the first half of the simulations, but transmission continues during the summer vacation. In model $R1$, the outbreak grows, albeit slowly, until day 200; in model $R2$, it peaks at day 100 (May 24) and wanes towards the end of the simulation. These patterns are consistent with the structure of the reference models: constant rates in model $R1$ and reduced transmission during vacation in model $R2$. The plots of the cumulative number of cases for the reference models in electronic supplementary material, figure S3 are also useful to visualize the continued/longer outbreak under the assumptions represented by these models, and therefore, their poor fit to the observed data. Finally, the case count in the full model after removing the intervention keeps increasing during the last weeks of the spring term until reaching a maximum of nearly 25 new cases per day at around day 100 (figure 3*b*), which is precisely when the summer vacation starts. The case count steadily decreases after that due to the reduced transmission rate during the summer. We interpret the peak as the effect of continuing with the original diagnosis procedure after day 61, since we set $q$ to 0 in the model.

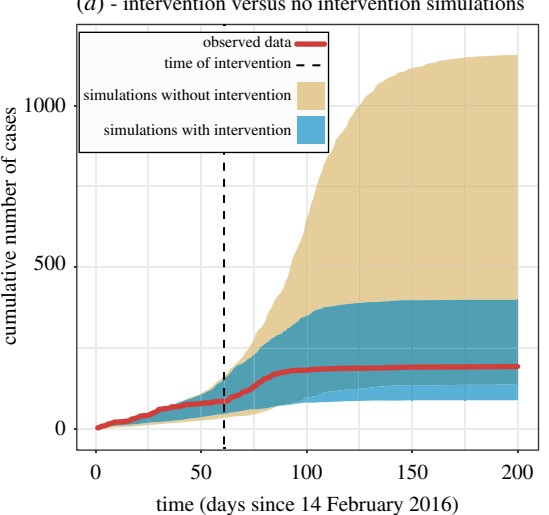

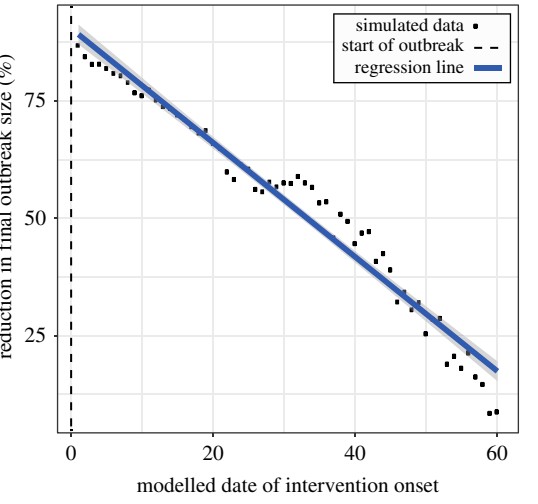

**Figure 4.** (*a*) Shows the comparison of the cumulative number of cases over time for the observed Harvard data and the range of cases (95% percentile of the runs) in simulations with and without interventions, with dotted lines representing the timing of the interventions. (*b*) The plot shows the percentage we expect the outbreak size to decrease by if the date of intervention had been moved up. There is a significant linear relationship between the time and percentage reduction.

## 3.3. Earlier intervention decreases outbreak size

The results from the intervention analysis for Harvard are depicted in figure 4. By the final day of the Harvard outbreak (day 130), the simulations without the intervention on day 61 yielded outbreak sizes that were up to four times the size of the actual outbreak (figure 4*a*). These results also indicate that the outbreak would have lasted much longer, if not for these vigilance-increasing strategies. By varying the day of the intervention from 1 to 61, we also obtained a linear regression between day of intervention and reduction of the outbreak (figure 4*b*). The simulated data show some random fluctuation due to the stochastic nature of the model, but the fitness of the regression is very high ($R^2 = 0.96$, $p < 10^{-9}$), and quick inspection of the plot suggests that if the new diagnosis protocol had been implemented within the first 10 days of the outbreak, then no more than 50 students would have been infected in total at Harvard.

# 4. Discussion

## 4.1. Effectiveness of HUHS interventions

The results from the likelihood ratio tests and simulated outbreak trajectories show that the full model, including relaxed interaction outside during school vacation and the effect of the updated diagnosis and isolation protocol, is the one that best describes the data from the 2016 mumps outbreak at Harvard University, among all alternative models considered in our analysis. Notably, the baseline removal rate $\gamma$ in this model is very high, suggesting that the initial diagnosis protocol was already quite effective at identifying and removing infected students from the population. This points to the effectiveness of the quarantine system implemented by HUHS. However, a small fraction of cases still managed to escape quarantine and keep the virus under circulation, as indicated by the reproduction number being higher than 1. Because of the model structure with two discrete transmission rates that alternate between school terms and breaks, the reproduction number goes below 1 during the spring break. This is reasonable given that most students are away due to the residential nature of the Harvard campus. Then, transmission increases again once the break is over. The implementation of the new diagnosis protocol on day 61, which required isolation if clinical symptoms were present, had a dramatic effect on the detection and isolation of positive cases, effectively taking the removal time to less than 1 day and the reproductive number below 0.6. Thanks to this key intervention, it was possible to end the outbreak before the beginning of the summer recess. The estimate of $\rho$ implies a remarkable reporting rate of 96%, suggesting that HUHS was able to identify most of the cases.

Reasons include the email awareness campaign, a community network—from resident deans to athletic coaches—reporting students and employees who seemed at-risk and more stringent diagnosis protocol, particularly towards the end of the outbreak.

The estimate for the overdispersion parameter $\psi$ is 0.54, suggesting that the actual data have more variability than expected under the assumed distribution. If $\psi$ had been approximately 0, the variance in our measurement model would have simplified to the variance for a binomial distribution. However, because the 95% CI is (0.22, 0.65) and thus does not include 0, we justify the modelling decision of representing the number of cases as an overdispersed binomial. This overdispersion in the observation model could be attributed to demographic and environmental stochasticity (e.g. a student during midterm season may be less likely to report symptoms), as well as the interventions themselves (e.g. reporting may increase temporarily after an awareness email) resulting in overdispersion in the number of reported cases.

## 4.2. Implications of intervention analysis

With the benefit of our intervention analysis, we conclude that the updated diagnosis protocol decreased the size of the Harvard outbreak by approximately three-fourths. Furthermore, for every day of intervention delay, we estimate that the outbreak size would have increased by 1.6% points, extrapolating the regression line in figure 4b. On the other hand, the results of the simulations using the full model with the $q$ parameter set 1, suggest that the outbreak would have been much larger if there was no change in the diagnosis protocols by April 15. According to these simulations, the total number of cases could have reached over one thousand, causing significant strain on the student healthcare system at Harvard and the increased probability for a fraction of students to experience complications from their infection.

Clearly, a limitation of this analysis is the assumption that everything remains the same while changing the time of the intervention under consideration. In reality, other factors might come into play if the outbreak becomes larger or smaller, which in turn could affect the dynamics of the outbreak as well as the interventions themselves. For instance, a dramatic increase in the number of cases as the result of a delayed intervention could incentivize students to adopt more proactive preventive measures to reduce their chances of becoming infected. Another limitation is that our model assumes that the effect of the HUHS intervention to be linear in its effect, i.e. the more aggressive the health workers were in case finding and quarantining symptomatic individuals, the more cases they prevented. However, there could be an inflection point where too aggressive of a measure could result in less case finding because of students decreased self-reporting as they may became apprehensive about being subjected to isolation and quarantine measures. While the high reporting rate in the fitted model suggests that this situation was unlikely at Harvard, more nuanced interventions could be represented within the POMP framework in order to represent such behavioural patterns.

Despite these limitations, our analysis provides a useful hypothetical quantification of the effect of accelerating or delaying interventions designed to contain the spread of an outbreak, and here, as expected, the sooner the interventions are introduced, the better the outcomes in terms of outbreak size. Of course, existing constraints in the school's health system could impede fast interventions. In such situations, our method can be useful to perform a cost–benefit analysis of how late an intervention could be made to still have a significant reduction in the health burden caused by the disease, with the added advantage of using a parsimonious model with a direct interpretation.

## 5. Conclusion

We constructed and parametrized a series of POMP models for the transmission of mumps on college campuses. POMP is a computationally efficient approach that accounts for the noisiness and incompleteness of case data. Moreover, it provides a flexible simulation framework to measure the effect of time-varying interventions as well as alternative outcome scenarios. Model selection is an important aspect in epidemiological analysis as well, and we showed how by defining a sequence of increasingly complex models, we can systematically test hypotheses of model fitness to the data. Given the deluge of modelling studies of COVID-19, rigorous model derivation and selection techniques are needed to ensure that models are a good fit to the available data and to quantify the effect of public health interventions under various scenarios.

While most literature today focuses on mumps prevention—such as administering third MMR doses to college-age students—this paper provides quantitative backing for more immediate and less costly approaches to mitigating the spread of mumps and other infectious diseases. Even with widespread availability of vaccines, outbreaks of highly transmissible diseases are still a reality, as mumps, and more recently COVID-19, illustrate very clearly. Quarantine protocols that incorporate epidemiological information to address false negatives in molecular tests could significantly reduce transmission and ultimately the size of outbreaks in environments with high levels of in-person interaction between individuals, such as college campuses.

# 6. Limitations

Some of our conclusions are likely affected by confounding factors that we cannot control for in this analysis. For example, the outbreak at Harvard started to subside in late April, not long before students finish the semester and leave campus, which would decrease the number of potential infections. The most promising method to determine the exact effect of isolation strategies is through a randomized control trial. We were fortunate to have direct access to school administrators who were involved in the response to the 2016 outbreak to discuss HUHS interventions in detail. More broadly, lack of publicly available datasets is a serious impediment to perform these kinds of analyses. Furthermore, detailed surveillance and diagnostic testing data, often very difficult to obtain, could complement basic epidemiological information, such as confirmed case counts, to provide a better understanding the prevalence of asymptomatic cases and false-positive rates and strengthen predictive analyses. Therefore, it will be essential that universities across the United States and the globe actively share data for comparative analysis, to identify the best intervention strategies to protect college campuses from outbreaks.

Ethics. Usage of Harvard University data for development of the SEIR model was approved by the Massachusetts Department of Public Health (MDPH) through protocol 906066. Harvard University Faculty of Arts and Sciences and the Broad Institute ceded review of secondary analysis to the MDPH IRB through institutional authorization agreements. The MDPH IRB waived informed consent given this research met the requirements pursuant to 45 CFR 46.116 (d).

Data accessibility. Data and relevant code for this research work are stored in GitHub https://github.com/colabobio/mumps-harvard-models and have been archived within the Zenodo repository https://zenodo.org/record/5740268. The data are provided in electronic supplementary material [32].

Authors' contributions. M.S.: conceptualization, data curation, formal analysis, methodology, software, visualization, writing—original draft, writing—review and editing; G.F.: methodology, software, writing—review and editing; S.F.: data curation, resources, supervision, writing—review and editing; P.J.B.: data curation, resources, supervision, writing—review and editing; P.C.S.: conceptualization, resources, supervision, writing—review and editing; A.C.: conceptualization, formal analysis, funding acquisition, methodology, project administration, resources, software, supervision, validation, visualization, writing—original draft, writing—review and editing. All authors gave final approval for publication and agreed to be held accountable for the work performed therein.

Competing interests. We declare no competing interests.

Funding. Howard Hughes Medical Institute, US National Institutes of Health.

Acknowledgements. The authors would like to thank Jonathan Grad and Joseph Lewnard for providing feedback on the study design, Hayden Metsky for reviewing the manuscript, members of MDPH for providing access to the Harvard data and Bridget Chak and Shirlee Wohl for guiding in the interpretation of the data.

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
