## [Peer Review File · Royal Society Open Science]

Review History

RSOS-210948.R0 (Original submission)

Review form: Reviewer 1

Is the manuscript scientifically sound in its present form?

No

Are the interpretations and conclusions justified by the results?

No

Is the language acceptable?

Yes

Do you have any ethical concerns with this paper?

No

Have you any concerns about statistical analyses in this paper?

No

Recommendation?

Major revision is needed (please make suggestions in comments)

Comments to the Author(s)

I reviewed this paper some months ago upon first submission and am revisiting it now. I appreciate the changes that the authors have made in dropping the OSU analysis which only raised questions and undermined conclusions from their more reliable Harvard data, as well as worked to test alternative hypotheses of the dynamics underlying the data. However, in this version of the paper, there are numerous typos, several missed removals of content from the earlier draft, as well as a few more serious concerns, which I highlight here:

1. Some parts of the paper have become much less organized upon resubmission and reframing. Currently, there are a host of headers and subheaders which are difficult to follow, and you seem to have skipped section 4.2 and repeated section 4.4 twice.

Right now the organization reads as:

1. Introduction. 2.2.1 Harvard mumps data. 2.1.2 interventions. 2.2 POMP model. 2.3.1. Process model. 2.3.2 Reference models. 2.3.3 Measurement model. 2.3.4 Final POMP models. 2.3 Fixed parameters. 2.4. MLE of free parameters. 2.5. Model selection by LRT. 2.6 intervention analysis description. 3.1.1. Likelihood of models and fits to the data. 3.2. Optimal parameter estimates. 3.2.1 ML estimates. 3.2.2. simulated outbreak trajectories. 3.3 Early intervention results. 4.1 Model selection and parameter interpretation. 4.3. Implications of intervention analysis. 4.4 Conclusions. 4.4 Limitations

I think that a large chunk of the methods could be moved to the supplementary material. Section 2.2 through section 2.6 could be reduced to one paragraph which states that you built an SEIR model in POMP, incorporating both process and observation models, and comparing a hypothesis of no constant transmission and recovery vs. a hypothesis of relaxed transmission outside of school terms vs. a third hypothesis of both relaxed transmission outside of school terms and an effective intervention. You then compared them using LRTs and AICs and simulated outbreaks under all model forms. The details of how this was all done can be moved to a supplementary doc to make a more readable paper. Certainly you do not need to include formulas for LRTs and AIC in the main text.

Likewise, sections 3.1.1 – 3.3 can be reduced to saying which model performed best and summarizing the main results of each figure. Details of AIC and LRT values can be kept to the tables (and table 1 and 2 could be combined!) and do not, in my opinion, need to be included in the text, especially since these numbers are meaningless outside of direct comparative context.

Finally, the first section of the discussion also includes much direct discussion of actual parameter estimates, material which is generally confined to the results section.

2. Susceptible population: I raised questions about the susceptible population assumptions in the first round of review, and I appreciate that consideration has been given to the international population on campus. In this reread, I am realizing that the authors model the entire population of Harvard, not just the undergraduate population. Is this an accurate rendition of the outbreak, or were cases largely restricted to undergrads? I wonder if the effective population size is actually smaller than represented here (i.e. corresponding to the undergraduate student body), even in the more conservative scenario.

3. Problems with R_e and R_0 : Following on #2, on line 248, the authors state that: $R_e(t) = \beta(t)/\gamma(t) * S(t)/N$, then further assume that “because $S(t) = N$, we can simplify this expression to $R_e(t) = \beta(t)/\gamma(t)$.”

This seems incorrect to me. $S(t) = N$ at $t=0$ where N is N_{eff} , the effective population size (this should be clarified), but $S(t)$ should decrease throughout the epidemic, thus decreasing R_e . In actuality, the authors are here approximating R_0 , not R_e , but doing so incorrectly. $R_0 = \beta/\gamma$ is the form of R_0 from an SIR model, but a Next Generation Matrix approach is needed for an SEIR model (see Hefferman et al 2005 or the very approachable “Notes on R_0 ” by James Holland Jones 2007 for an explanation).

To obtain R_e , the correct estimate of R_0 (which will time-vary as β and γ vary in the interaction model) should then be multiplied by $S(t)$ at the timepoint of the epidemic.

Alternatively, R_e can also be estimated from the simulated case data, using any number of methods reviewed in Gostic et al. 2020.

4. Finally, by setting the population size to N_{eff} , a number corresponding to the waned immunity population, the authors ignore the fact that half the population is still vaccinated and mixed in with these contacts. If modeled explicitly, this would likely alter dynamics and parameter estimates, as some transmissions would be “wasted” on dead-end hosts (e.g. frequency-dependent transmission). At a minimum, this should be considered in the discussion.

Additional line-by-line comments here – there are many typos in the current version of the paper:

L27: COVID-19 comment comes out of the blue. Needs a bit more context. For example, try: “As is illustrated in the current COVID-19 pandemic...”

L114: typo: “because the of its transient replication”

L115: I appreciate the addition of information as to why mumps is hard to test for via PCR but I would recommend dropping the second clause about the “concomitant presence of antibodies.” The referenced paper suggests this may increase the difficulty of isolating mumps but likely won’t impact PCR-based detection.

L216: missing ‘the’ pathogen’s infectivity

L217: β captures (a) transmissibility of the infectious host, (b) susceptibility of the recipient host, and (c) contact rates between the two if you want to be truly accurate

L233: The link in the reference provided (21) fails, and it does not appear to be one of the many seminal TSIR papers on measles and school terms out there. Try Grenfell et al 2002 or Bjornstad et al 2002

L248: I am confused how $S(t) = N$. See comments above in #3.

L299: See comments above about Harvard population size under #2.

L367: It would be helpful to refer to these models by their actual intervention, rather than the obscure names I0, I1, I2, which have not been explained.

L389: change ‘indicating’ to ‘suggesting’

L432: “are” depicted

L490: give some of these examples

L 525: you mention OSU here, clearly not edited from the first draft, as it has now been dropped.

Fig 2 caption: As in L367, it would be helpful to name the reference models and what their function was rather than just numbering them here. That said, I don’t think it is necessary to include poorly fitted models in the actual paper – you could move them to the SI and just keep panel A and B.

Fig 3 caption: The caption refers to panel C which does not exist. Additionally, I suggest relabelling the x-axis on panel B to be “modeled date of intervention onset” and label a bar on the left side with “start of epidemic”

Review form: Reviewer 2

Is the manuscript scientifically sound in its present form?

Yes

Are the interpretations and conclusions justified by the results?

Yes

Is the language acceptable?

Yes

Do you have any ethical concerns with this paper?

No

Have you any concerns about statistical analyses in this paper?

No

Recommendation?

Accept with minor revision (please list in comments)

Comments to the Author(s)

Thank you for the opportunity to review "Containing Mumps Spread on College Campus" by Shah and colleagues. In their analysis, the authors model the impact of mitigation strategies for controlling mumps transmission on a college campus. They use data from the 2016 outbreak of Mumps at Harvard University and Greater Boston area, which involved 210 confirmed cases, to fit SEIR model parameters. They focused on assessing the impact to two main interventions: 1.) Aggressive quarantine measure based on symptoms alone the absence of a diagnostic test, and 2.) Strict isolation of a suspected cases. To test the robustness of their analysis, they included three models of increasing complexity, finding that the "full intervention" model including the updated diagnosis protocol best fit the data. Most significantly, they find that the updated diagnosis protocol decreased the size of the Harvard outbreak by approximately three-fourths, which reinforces the notion that preventing cases early on pays dividends later in the epidemics. Overall, I find their model straightforward, parsimonious, and easy to follow. Further, the inclusion of the two simpler models made the results for the full model more compelling. I feel that the results are of significant interest, especially in the post-COVID era as they can help administrators make hard decisions about stronger mitigation strategies earlier on in an epidemic response. I only have a few general questions and discussion points as well as some minor comments.

Regarding case definitions (line 188), it states that confirmed cases are those with a positive laboratory test for mumps virus, and probable cases are those who either tested positive for the anti-mumps IgM antibody or had an epidemiologic linkage to another probable or confirmed case. I feel that the "suspected case" definition used in the "updated diagnosis protocol" should be added to this as well, and it should be stated what "clinical symptoms of mumps" were used to identify them. I am also interested to know if false positives from "suspected cases" were ever quantified/estimated through follow-up serological testing. Depending on the disease and the severity of the intervention, the false positive rate should definitely be weighed in the context of false negatives.

One interesting point to consider is that you assume the parameter q to be generally linear in its effect i.e., the more aggressive you are in case finding and quarantining symptomatic individuals, the more cases you prevent. However, there could be an inflection point where too aggressive of a measure could result in less case finding because of students decreased self-reporting. I feel this

phenomenon was common during the COVID pandemic as individuals were apprehensive about being subjected to isolation and quarantine measures, which translates into financial loss. During the mumps outbreak, were there any anecdotes or evidence that students were apprehensive about reporting their symptoms? Perhaps instead, individuals were self-isolating. I am wondering what impact this would have in your model, although it may be hard to determine the time to “breaking point” when the intervention seemingly backfires. Similarly, was there any indication that students were getting boosters on their own and possibly reducing the pool of susceptibles?

Minor:

Line 114 – errant “the”

Line 224 – I had a feeling based on experiences managing COVID on a university campus that emails would not be very effective. Is there any empiric data to support this, possibly in the form of a survey?

Can the MMR vaccine be used as post-exposure prophylaxis??

Decision letter (RSOS-210948.R0)

Dear Dr Colubri

The Editors assigned to your paper RSOS-210948 "Containing the Spread of Mumps on College Campuses" have now received comments from reviewers and would like you to revise the paper in accordance with the reviewer comments and any comments from the Editors. Please note this decision does not guarantee eventual acceptance.

Please submit your revised manuscript and required files (see below) no later than 21 days from today's (ie 18-Oct-2021) date. Note: the ScholarOne system will 'lock' if submission of the revision is attempted 21 or more days after the deadline. If you do not think you will be able to meet this deadline please contact the editorial office immediately.

Please note article processing charges apply to papers accepted for publication in Royal Society Open Science (<https://royalsocietypublishing.org/rsos/charges>). Charges will also apply to papers transferred to the journal from other Royal Society Publishing journals, as well as papers submitted as part of our collaboration with the Royal Society of Chemistry

(<https://royalsocietypublishing.org/rsos/chemistry>). Fee waivers are available but must be requested when you submit your revision (<https://royalsocietypublishing.org/rsos/waivers>).

on behalf of Prof Marta Kwiatkowska (Subject Editor)
openscience@royalsociety.org

Associate Editor Comments to Author:

Please accept our apologies for the delay in completing the review of your work - an unusually large number of invitations were issued before we were able to secure the kind agreement of the reviewers here. Given the comments received, which are substantial, but appear eminently achievable, we would ask that you revise the manuscript to address the concerns raised. The reviewers will be asked to take a further look at your revised manuscript in due course - if they are satisfied with the revisions made, we hope we may accept the paper; however, if they continue to express concerns, we may not be able to consider the paper further. Good luck and we'll look forward to receiving the revision.

Reviewer comments to Author:

Reviewer: 1

Comments to the Author(s)

I reviewed this paper some months ago upon first submission and am revisiting it now. I appreciate the changes that the authors have made in dropping the OSU analysis which only raised questions and undermined conclusions from their more reliable Harvard data, as well as worked to test alternative hypotheses of the dynamics underlying the data. However, in this version of the paper, there are numerous typos, several missed removals of content from the earlier draft, as well as a few more serious concerns, which I highlight here:

1. Some parts of the paper have become much less organized upon resubmission and reframing. Currently, there are a host of headers and subheaders which are difficult to follow, and you seem to have skipped section 4.2 and repeated section 4.4 twice.

Right now the organization reads as:

1. Introduction. 2.2.1 Harvard mumps data. 2.1.2 interventions. 2.2 POMP model. 2.3.1. Process model. 2.3.2 Reference models. 2.3.3 Measurement model. 2.3.4 Final POMP models. 2.3 Fixed parameters. 2.4. MLE of free parameters. 2.5. Model selection by LRT. 2.6 intervention analysis description. 3.1.1. Likelihood of models and fits to the data. 3.2. Optimal parameter estimates. 3.2.1 ML estimates. 3.2.2. simulated outbreak trajectories. 3.3 Early intervention results. 4.1 Model selection and parameter interpretation. 4.3. Implications of intervention analysis. 4.4 Conclusions. 4.4 Limitations

I think that a large chunk of the methods could be moved to the supplementary material. Section 2.2 through section 2.6 could be reduced to one paragraph which states that you built an SEIR model in POMP, incorporating both process and observation models, and comparing a

hypothesis of no constant transmission and recovery vs. a hypothesis of relaxed transmission outside of school terms vs. a third hypothesis of both relaxed transmission outside of school terms and an effective intervention. You then compared them using LRTs and AICs and simulated outbreaks under all model forms. The details of how this was all done can be moved to a supplementary doc to make a more readable paper. Certainly you do not need to include formulas for LRTs and AIC in the main text.

Likewise, sections 3.1.1–3.3 can be reduced to saying which model performed best and summarizing the main results of each figure. Details of AIC and LRT values can be kept to the tables (and table 1 and 2 could be combined!) and do not, in my opinion, need to be included in the text, especially since these numbers are meaningless outside of direct comparative context.

Finally, the first section of the discussion also includes much direct discussion of actual parameter estimates, material which is generally confined to the results section.

2. Susceptible population: I raised questions about the susceptible population assumptions in the first round of review, and I appreciate that consideration has been given to the international population on campus. In this reread, I am realizing that the authors model the entire population of Harvard, not just the undergraduate population. Is this an accurate rendition of the outbreak, or were cases largely restricted to undergrads? I wonder if the effective population size is actually smaller than represented here (i.e. corresponding to the undergraduate student body), even in the more conservative scenario.

3. Problems with R_e and R_0 : Following on #2, on line 248, the authors state that: $R_e(t) = \beta(t)/\gamma(t) * S(t)/N$, then further assume that “because $S(t) = N$, we can simplify this expression to $R_e(t) = \beta(t)/\gamma(t)$.”

This seems incorrect to me. $S(t) = N$ at $t=0$ where N is N_{eff} , the effective population size (this should be clarified), but $S(t)$ should decrease throughout the epidemic, thus decreasing R_e . In actuality, the authors are here approximating R_0 , not R_e , but doing so incorrectly. $R_0 = \beta/\gamma$ is the form of R_0 from an SIR model, but a Next Generation Matrix approach is needed for an SEIR model (see Heffernan et al 2005 or the very approachable “Notes on R_0 ” by James Holland Jones 2007 for an explanation).

To obtain R_e , the correct estimate of R_0 (which will time-vary as β and γ vary in the interaction model) should then be multiplied by $S(t)$ at the timepoint of the epidemic. Alternatively, R_e can also be estimated from the simulated case data, using any number of methods reviewed in Gostic et al. 2020.

4. Finally, by setting the population size to N_{eff} , a number corresponding to the waned immunity population, the authors ignore the fact that half the population is still vaccinated and mixed in with these contacts. If modeled explicitly, this would likely alter dynamics and parameter estimates, as some transmissions would be “wasted” on dead-end hosts (e.g. frequency-dependent transmission). At a minimum, this should be considered in the discussion.

Additional line-by-line comments here – there are many typos in the current version of the paper:

L27: COVID-19 comment comes out of the blue. Needs a bit more context. For example, try: “As is illustrated in the current COVID-19 pandemic...”

L114: typo: “because the of its transient replication”

L115: I appreciate the addition of information as to why mumps is hard to test for via PCR but I would recommend dropping the second clause about the “concomitant presence of antibodies.” The referenced paper suggests this may increase the difficulty of isolating mumps but likely won’t impact PCR-based detection.

L216: missing ‘the’ pathogen’s infectivity

L217: beta captures (a) transmissibility of the infectious host, (b) susceptibility of the recipient host, and (c) contact rates between the two if you want to be truly accurate

L233: The link in the reference provided (21) fails, and it does not appear to be one of the many seminal TSIR papers on measles and school terms out there. Try Grenfell et al 2002 or Bjornstad et al 2002

L248: I am confused how $S(t) = N$. See comments above in #3.

L299: See comments above about Harvard population size under #2.

L367: It would be helpful to refer to these models by their actual intervention, rather than the obscure names I0, I1, I2, which have not been explained.

L389: change 'indicating' to 'suggesting'

L432: "are" depicted

L490: give some of these examples

L 525: you mention OSU here, clearly not edited from the first draft, as it has now been dropped.

Fig 2 caption: As in L367, it would be helpful to name the reference models and what their function was rather than just numbering them here. That said, I don't think it is necessary to include poorly fitted models in the actual paper - you could move them to the SI and just keep panel A and B.

Fig 3 caption: The caption refers to panel C which does not exist. Additionally, I suggest relabelling the x-axis on panel B to be "modeled date of intervention onset" and label a bar on the left side with "start of epidemic"

Reviewer: 2

Comments to the Author(s)

Thank you for the opportunity to review "Containing Mumps Spread on College Campus" by Shah and colleagues. In their analysis, the authors model the impact of mitigation strategies for controlling mumps transmission on a college campus. They use data from the 2016 outbreak of Mumps at Harvard University and Greater Boston area, which involved 210 confirmed cases, to fit SEIR model parameters. They focused on assessing the impact to two main interventions: 1.) Aggressive quarantine measure based on symptoms alone the absence of a diagnostic test, and 2.) Strict isolation of a suspected cases. To test the robustness of their analysis, they included three models of increasing complexity, finding that the "full intervention" model including the updated diagnosis protocol best fit the data. Most significantly, they find that the updated diagnosis protocol decreased the size of the Harvard outbreak by approximately three-fourths, which reinforces the notion that preventing cases early on pays dividends later in the epidemics. Overall, I find their model straightforward, parsimonious, and easy to follow. Further, the inclusion of the two simpler models made the results for the full model more compelling. I feel that the results are of significant interest, especially in the post-COVID era as they can help administrators make hard decisions about stronger mitigation strategies earlier on in an epidemic response. I only have a few general questions and discussion points as well as some minor comments.

Regarding case definitions (line 188), it states that confirmed cases are those with a positive laboratory test for mumps virus, and probable cases are those who either tested positive for the anti-mumps IgM antibody or had an epidemiologic linkage to another probable or confirmed case. I feel that the "suspected case" definition used in the "updated diagnosis protocol" should be added to this as well, and it should be stated what "clinical symptoms of mumps" were used to identify them. I am also interested to know if false positives from "suspected cases" were ever quantified/estimated through follow-up serological testing. Depending on the disease and the severity of the intervention, the false positive rate should definitely be weighed in the context of false negatives.

One interesting point to consider is that you assume the parameter q to be generally linear in its effect i.e., the more aggressive you are in case finding and quarantining symptomatic individuals, the more cases you prevent. However, there could be an inflection point where too aggressive of a measure could result in less case finding because of students decreased self-reporting. I feel this phenomenon was common during the COVID pandemic as individuals were apprehensive about being subjected to isolation and quarantine measures, which translates into financial loss. During the mumps outbreak, were there any anecdotes or evidence that students were apprehensive about reporting their symptoms? Perhaps instead, individuals were self-isolating. I am wondering what impact this would have in your model, although it may be hard to determine the time to “breaking point” when the intervention seemingly backfires. Similarly, was there any indication that students were getting boosters on their own and possibly reducing the pool of susceptibles?

Minor:

Line 114 - errant “the”

Line 224 - I had a feeling based on experiences managing COVID on a university campus that emails would not be very effective. Is there any empiric data to support this, possibly in the form of a survey?

Can the MMR vaccine be used as post-exposure prophylaxis??

===PREPARING YOUR MANUSCRIPT===

===PREPARING YOUR REVISION IN SCHOLARONE===

Author's Response to Decision Letter for (RSOS-210948.R0)

See Appendix A.

RSOS-210948.R1 (Revision)

Review form: Reviewer 1

Is the manuscript scientifically sound in its present form?

Yes

Are the interpretations and conclusions justified by the results?

Yes

Is the language acceptable?

Yes

Do you have any ethical concerns with this paper?

No

Have you any concerns about statistical analyses in this paper?

No

Recommendation?

Accept with minor revision (please list in comments)

Comments to the Author(s)

This is my third review of this paper. It has been enjoyable to observe its progression and improvement over the past year, and I thank the authors for their edits. I am now satisfied with the edits the others have made to the manuscript – most especially moving the details of the methods to the supplementary materials, the tightening of the manuscript, the addition of the very helpful Fig2, and the thorough address of the effective population size and assumptions related to R0 and Re. I have just a few minor comments that can be easily addressed below.

Also, I'll note that the 'response to reviewers' refers to R1 as a 'he' but I am a 'she'. Best not to make assumptions!

Minor comments:

Line 86: "confirmed cases are those with a positive laboratory test for mumps virus." – is this always a PCR test? If so, please specify

Line 111: you say that the second intervention was to "treat and isolate" anyone suspected of mumps but this seems to be more intervention #3. It seems to me that there are three

interventions, which can be grouped as (1) awareness raising, (2) case identification and testing, (3) isolation of infectious individuals. I recommend you edit this intro section to reflect this.

Line 187: It is fine to use a larger population size including undergrads, grad students, etc, but you do specify that the 53% susceptible corresponds to 20-29 year olds. Presumably your total population includes individuals above (staff) and below (undergrads) that age range – is the 53% still valid? Just an acknowledgement of the limits of this assumption would be fine

Line 247: typo: “a the lower”

Line 271: But isn't this assuming that the population size remains constant? And students would typically leave for the summer, so you'd likely get an end to the local epidemic anyhow, wouldn't you? It should be mentioned here that the projected epidemic without intervention is under assumptions in which the population size is not changed.

Line 285-287: I suggest changing this to say that “the plot SUGGESTS that if the new diagnosis protocol had been implemented within the first 10 days of the outbreak...” rather than ‘reveals’ as we will never really know what might have happened

Line 391: I would remove mention of the “post-COVID-19” world as it does not seem like that is ever going to be a reality at this point

Supplement :

Line 178: “times of the values” should be “times the values”

Line 188: “two-doses mumps” should be “two-dose mumps”

Line 226: “three model” should be “models”

Review form: Reviewer 2

Is the manuscript scientifically sound in its present form?

Yes

Are the interpretations and conclusions justified by the results?

Yes

Is the language acceptable?

Yes

Do you have any ethical concerns with this paper?

No

Have you any concerns about statistical analyses in this paper?

No

Recommendation?

Accept as is

Comments to the Author(s)

Thank you for the opportunity to review the revised manuscript “Containing the Spread of Mumps on College Campuses” by Mirai and colleagues. The authors sufficiently addressed my epidemiological and translational questions I posed in my primary review. I also appreciated the discussion generated by the comments from Reviewer 1 regarding the model and the layout of the manuscript. Restructuring of the manuscript and moving the methods to the supplementary material significantly improved the readability. The addition of Figure 2 that details the model was also helpful for easy reference. I have not additional comments at this time.

Decision letter (RSOS-210948.R1)

Dear Dr Colubri

On behalf of the Editors, we are pleased to inform you that your Manuscript RSOS-210948.R1 "Containing the Spread of Mumps on College Campuses" has been accepted for publication in Royal Society Open Science subject to minor revision in accordance with the referees' reports. Please find the referees' comments along with any feedback from the Editors below my signature.

Please submit your revised manuscript and required files (see below) no later than Friday 14 January 2022. Note: the ScholarOne system will 'lock' if submission of the revision is attempted after the deadline. If you do not think you will be able to meet this deadline please contact the editorial office immediately.

on behalf of Prof Marta Kwiatkowska (Subject Editor)
openscience@royalsociety.org

Associate Editor Comments to Author:

The reviewers have a handful of remaining comments - please aim to address these in your final revision. Given the imminent festive break, the editorial team will set the deadline for revision into the New Year.

Reviewer comments to Author:

Reviewer: 2

Comments to the Author(s)

Thank you for the opportunity to review the revised manuscript "Containing the Spread of Mumps on College Campuses" by Mirai and colleagues. The authors sufficiently addressed my

epidemiological and translational questions I posed in my primary review. I also appreciated the discussion generated by the comments from Reviewer 1 regarding the model and the layout of the manuscript. Restructuring of the manuscript and moving the methods to the supplementary material significantly improved the readability. The addition of Figure 2 that details the model was also helpful for easy reference. I have not additional comments at this time.

Reviewer: 1

Comments to the Author(s)

This is my third review of this paper. It has been enjoyable to observe its progression and improvement over the past year, and I thank the authors for their edits. I am now satisfied with the edits the others have made to the manuscript – most especially moving the details of the methods to the supplementary materials, the tightening of the manuscript, the addition of the very helpful Fig2, and the thorough address of the effective population size and assumptions related to R_0 and R_e . I have just a few minor comments that can be easily addressed below.

Also, I'll note that the 'response to reviewers' refers to R1 as a 'he' but I am a 'she'. Best not to make assumptions!

Minor comments:

Line 86: "confirmed cases are those with a positive laboratory test for mumps virus." – is this always a PCR test? If so, please specify

Line 111: you say that the second intervention was to "treat and isolate" anyone suspected of mumps but this seems to be more intervention #3. It seems to me that there are three interventions, which can be grouped as (1) awareness raising, (2) case identification and testing, (3) isolation of infectious individuals. I recommend you edit this intro section to reflect this.

Line 187: It is fine to use a larger population size including undergrads, grad students, etc, but you do specify that the 53% susceptible corresponds to 20-29 year olds. Presumably your total population includes individuals above (staff) and below (undergrads) that age range – is the 53% still valid? Just an acknowledgement of the limits of this assumption would be fine

Line 247: typo: "a the lower"

Line 271: But isn't this assuming that the population size remains constant? And students would typically leave for the summer, so you'd likely get an end to the local epidemic anyhow, wouldn't you? It should be mentioned here that the projected epidemic without intervention is under assumptions in which the population size is not changed.

Line 285-287: I suggest changing this to say that "the plot SUGGESTS that if the new diagnosis protocol had been implemented within the first 10 days of the outbreak..." rather than 'reveals' as we will never really know what might have happened

Line 391: I would remove mention of the "post-COVID-19" world as it does not seem like that is ever going to be a reality at this point

Supplement :

Line 178: "times of the values" should be "times the values"

Line 188: "two-doses mumps" should be "two-dose mumps"

Line 226: "three model" should be "models"

===PREPARING YOUR MANUSCRIPT===

one version should clearly identify all the changes that have been made (for instance, in coloured highlight, in bold text, or tracked changes);
 a 'clean' version of the new manuscript that incorporates the changes made, but does not highlight them. This version will be used for typesetting.

===PREPARING YOUR REVISION IN SCHOLARONE===

-- If you are requesting an article processing charge waiver, you must select the relevant waiver option (if requesting a discretionary waiver, the form should have been uploaded, see 'File upload' above).

-- If you have uploaded any electronic supplementary (ESM) files, please ensure you follow the guidance at <https://royalsociety.org/journals/authors/author-guidelines/#supplementary-material> to include a suitable title and informative caption. An example of appropriate titling and captioning may be found at https://figshare.com/articles/Table_S2_from_Is_there_a_trade-off_between_peak_performance_and_performance_breadth_across_temperatures_for_aerobic_scope_in_teleost_fishes_/3843624.

Author's Response to Decision Letter for (RSOS-210948.R1)

See Appendix B.

Decision letter (RSOS-210948.R2)

Dear Dr Colubri,

I am pleased to inform you that your manuscript entitled "Containing the Spread of Mumps on College Campuses" is now accepted for publication in Royal Society Open Science.

on behalf of Prof Marta Kwiatkowska (Subject Editor)
openscience@royalsociety.org

Appendix A

Dear RSOS Editors,

We are very grateful that we had the opportunity to address the reviewers' comments of our manuscript "Containing the Spread of Mumps on College Campuses". Both reviewers are positive about our modeling approach and usefulness of the conclusions, but also suggested several improvements in the text and figures. Thanks to these insightful comments, we feel that we have arrived at a significantly improved manuscript that offers a meaningful contribution to the epidemiological research of infectious disease outbreaks on college campuses. Our current reality in the times of COVID-19 makes this type of work particularly relevant.

Best regards,
The Authors

Reviewer 1

I reviewed this paper some months ago upon first submission and am revisiting it now. I appreciate the changes that the authors have made in dropping the OSU analysis which only raised questions and undermined conclusions from their more reliable Harvard data, as well as worked to test alternative hypotheses of the dynamics underlying the data. However, in this version of the paper, there are numerous typos, several missed removals of content from the earlier draft, as well as a few more serious concerns, which I highlight here:

We are glad that reviewer #1, who reviewed our original submission to JRSI, was able to review this updated submission and follow up on his initial comments. Reviewer #1 read (both) manuscripts with great attention to detail, and in addressing the comments below we realized several ways in which our manuscript could be improved.

1. Some parts of the paper have become much less organized upon resubmission and reframing. Currently, there are a host of headers and subheaders which are difficult to follow, and you seem to have skipped section 4.2 and repeated section 4.4 twice.

We did a substantial reorganization of the manuscript following this and the comment below, and updated/simplified the headers and section numbering accordingly.

Right now the organization reads as:

1. Introduction. 2.2.1 Harvard mumps data. 2.1.2 interventions. 2.2 POMP model. 2.3.1. Process model. 2.3.2 Reference models. 2.3.3 Measurement model. 2.3.4 Final POMP models. 2.3 Fixed parameters. 2.4. MLE of free parameters. 2.5. Model selection by LRT. 2.6 intervention analysis description. 3.1.1. Likelihood of models and fits to the data. 3.2. Optimal parameter estimates. 3.2.1 ML estimates. 3.2.2. simulated outbreak trajectories. 3.3 Early intervention results. 4.1 Model selection and parameter interpretation. 4.3. Implications of intervention analysis. 4.4 Conclusions. 4.4 Limitations

I think that a large chunk of the methods could be moved to the supplementary material. Section 2.2 through section 2.6 could be reduced to one paragraph which states that you built an SEIR model in POMP, incorporating both process and observation models, and comparing a hypothesis of no constant transmission and recovery vs. a hypothesis of relaxed transmission outside of school terms vs. a third hypothesis of both relaxed transmission outside of school terms and an effective intervention. You then compared them using LRTs and AICs and simulated outbreaks under all model forms. The details of how this was all done can be moved to a supplementary doc to make a more readable paper. Certainly you do not need to include formulas for LRTs and AIC in the main text.

We completely agree about moving most of the details in the methods to the supplementary materials, and we have done so. We wrote an overview of the methods in the main text (section 2.3), which extends over three paragraphs, but is still much more concise than the original methods description, which is now available in the supplementary document for those readers interested in the fine details. We also added a new figure to the main text (Figure 2) showing a schematic representation of the SEIR model, and the three parametrizations we considered in our analysis. We hope this figure helps orient the reader and serves as a visual reference regarding the three models described in the text.

Likewise, sections 3.1.1—3.3 can be reduced to saying which model performed best and summarizing the main results of each figure. Details of AIC and LRT values can be kept to the tables (and table 1 and 2 could be combined!) and do not, in my opinion, need to be included in the text, especially since these numbers are meaningless outside of direct comparative context.

We moved the details about AIC and LRT to the supplementary materials (Section S1.4) and combined tables 1 and 2 in the main text. This makes much easier to quickly glance at the model parameters.

Finally, the first section of the discussion also includes much direct discussion of actual parameter estimates, material which is generally confined to the results section.

We moved the parts of text previously in the results that were actually discussing the estimates into the discussion section and left only the presentation of the estimates in the results section.

2. Susceptible population: I raised questions about the susceptible population assumptions in the first round of review, and I appreciate that consideration has been given to the international population on campus. In this reread, I am realizing that the authors model the entire population of Harvard, not just the undergraduate population. Is this an accurate rendition of the outbreak, or were cases largely restricted to undergrads? I wonder if the effective population size is actually smaller than represented here (i.e. corresponding to the undergraduate student body), even in the more conservative scenario.

This is a great point, and we addressed it in the main text in detail (L188-191). Essentially, the argument is that, even though the outbreak mostly affected undergraduate students, it spread widely among the school community as discussed in Wohl et al., with some members of the university staff becoming infected. Although the majority of cases occurred among the students in Harvard College (which also comprises the larger subgroup of the population at Harvard) this evidence makes us confident in using the entire population in our models.

3. Problems with R_e and R_0 : Following on #2, on line 248, the authors state that: $R_e(t) = \beta(t)/\gamma(t) * S(t)/N$, then further assume that “because $S(t) = N$, we can simplify this expression to $R_e(t) = \beta(t)/\gamma(t)$.”

This seems incorrect to me. $S(t) = N$ at $t=0$ where N is N_{eff} , the effective population size (this should be clarified), but $S(t)$ should decrease throughout the epidemic, thus decreasing R_e . In actuality, the authors are here approximating R_0 , not R_e , but doing so incorrectly. $R_0 = \beta/\gamma$ is the form of R_0 from an SIR model, but a Next Generation Matrix approach is needed for an SEIR model (see Heffernan et al 2005 or the very approachable “Notes on R_0 ” by James Holland Jones 2007 for an explanation).

To obtain R_e , the correct estimate of R_0 (which will time-vary as β and γ vary in the interaction model) should then be multiplied by $S(t)$ at the timepoint of the epidemic.

Alternatively, R_e can also be estimated from the simulated case data, using any number of methods reviewed in Gostic et al. 2020.

We thank the reviewer for pointing out to the potential issues with R_0 and R_e and for providing the additional materials. This prompted us to review the literature in more detail and ascertain whether our approximations are valid or need to be corrected. With regards to R_0 , while it is true that the general formula for SEIR models is not what we used, it turns out that under the assumption of no susceptible students joining or leaving the population during the outbreak (noted in L165 in the main text, and in more depth in the supplementary L100-103) the formula reduces to $R_0 = \beta/\gamma$. Regarding R_e , Gostic provides the general definition $R_e = \beta/\gamma * S(t)/N$. If $S(t)/N$ is close to 1, then our approximation for R_e would also be justifiable, as long as the quantitative results (R_e greater or smaller than 1 at various points of the outbreak) can still be inferred. This is indeed the case, to demonstrate we constructed an interval that contains R_e with boundaries $\beta/\gamma * S_{min}/N$ and β/γ , where S_{min} is the minimum number of susceptibles calculated over all our simulated trajectories (using the MLEs). This interval turns out to be quite narrow, so qualitative results are not affected. This is described in detail in the supplementary in SL103-108 and more briefly in the main text (L240-243).

4. Finally, by setting the population size to N_{eff} , a number corresponding to the waned immunity population, the authors ignore the fact that half the population is still vaccinated and mixed in with these contacts. If modeled explicitly, this would likely alter dynamics and parameter estimates, as some transmissions would be “wasted” on dead-end hosts (e.g. frequency-dependent transmission). At a minimum, this should be considered in the discussion.

This is a very interesting point, and after careful consideration of the meaning of the transmission rates in compartmental models in general, we concluded that the dynamics we obtain from our POMP models are still valid, but the “effective” beta coefficient needs to be interpreted with some care. We reproduce our reasoning provided in the supplementary materials (SL167-178):

“This modeling decision of using N_{eff} would not have an effect in the simulated epidemic dynamics, as contacts between susceptible and immune students do not result in any changes in actual transmissions and recoveries. However, we should note that the use of the effective population size has an implication with regards to the interpretation of the baseline transmission rate β . This rate is defined, without any loss of generality, as the product between the population contact rate and the probability of an infected individual of passing the disease to a susceptible during a contact event, in other words: $\beta=c \times v$, where v is the probability of infection (12). Since the effect of considering an effective population of susceptible individuals surrounded by a significant proportion of immune individuals is a smaller “effective” contact rate (as our model only “cares” about contacts between susceptibles), we could conclude that the contact rate among all members of the population, and thus the transmission rate β if all students were susceptible, is precisely 0.53 times of the values estimated from the “effective” population.”

We did look into the term the reviewer suggested (frequency-dependent transmission) but after going through the literature (for example Begon et al. (2002) A clarification of transmission terms in host-microparasite models: numbers, densities and areas, and this blog post: <https://parasiteecology.wordpress.com/2013/10/17/density-dependent-vs-frequency-dependent-disease-transmission/>) we think this would not affect our models. The transmission of mumps is clearly density-dependent, but since we are in a constant population and area scenario, the density -dependent rates are equivalent to the frequency-dependent rates, as far as we understand it. Regarding the observation that transmissions to immune students would be “wasted”, that makes intuitive sense, but we think this is not affecting the dynamics per-se, but the meaning of the transmission rate, and more specifically, the contact rate, as noted above. But we are happy to continue this discussion with the reviewer if needed.

Additional line-by-line comments here—there are many typos in the current version of the paper:

We appreciate the effort the reviewer put in pointing out typos, which were indeed many, and providing line-by-line comments. We address them individually below.

L27: COVID-19 comment comes out of the blue. Needs a bit more context. For example, try: “As is illustrated in the current COVID-19 pandemic...”

Added the suggested opening for that sentence (L27)

L115: I appreciate the addition of information as to why mumps is hard to test for via PCR but I would recommend dropping the second clause about the “concomitant presence of antibodies.” The referenced paper suggests this may increase the difficulty of isolating mumps but likely won’t impact PCR-based detection.

Dropped the second clause (L119).

L217: beta captures (a) transmissibility of the infectious host, (b) susceptibility of the recipient host, and (c) contact rates between the two if you want to be truly accurate

Clarified in the supplementary materials, see lines SL69-70.

L233: The link in the reference provided (21) fails, and it does not appear to be one of the many seminal TSIR papers on measles and school terms out there. Try Grenfell et al 2002 or Bjornstad et al 2002

Replaced with the Grenfell et al reference, as suggested.

L248: I am confused how $S(t) = N$. See comments above in #3.

We clarified in the discussion of the approximation for R_e , as mentioned above. See lines L240-244 in main text, lines SL100-108 in the supplementary.

L299: See comments above about Harvard population size under #2.

This has been addressed as well, see lines L187-192 in the main text.

L367: It would be helpful to refer to these models by their actual intervention, rather than the obscure names I0, I1, I2, which have not been explained.

This is a great suggestion, we renamed the reference models to R1 and R2, and the intervention model to “full” or F since it includes both the effect of vacations and the HUHS intervention. Together with the new Figure 2, we believe it will be much easier for the readers to differentiate among the different models.

L490: give some of these examples

We added one example of how interventions could alter the dynamics of an outbreak (L339), obviously there are many more but this one is particularly relevant to the context of a college campus.

Fig 2 caption: As in L367, it would be helpful to name the reference models and what their function was rather than just numbering them here. That said, I don't think it is necessary to include poorly fitted models in the actual paper – you could move them to the SI and just keep panel A and B.

We moved panels C & D in Figure 2 (now 3) to supplementary figure 2, this greatly helps with readability.

Fig 3 caption: The caption refers to panel C which does not exist. Additionally, I suggest relabelling the x-axis on panel B to be “modeled date of intervention onset” and label a bar on the left side with “start of epidemic”

We made the suggested changes and corrections in figure 3 (now 4). We also added an inset with some extra information in panel B.

L114: typo: “because the of its transient replication”

L216: missing ‘the’ pathogen’s infectivity

L389: change ‘indicating’ to ‘suggesting’

L432: “are” depicted

L 525: you mention OSU here, clearly not edited from the first draft, as it has now been dropped.

Thanks for pointing out to these typos, all have been corrected.

Reviewer 2

Thank you for the opportunity to review “Containing Mumps Spread on College Campus” by Shah and colleagues. In their analysis, the authors model the impact of mitigation strategies for controlling mumps transmission on a college campus. They use data from the 2016 outbreak of Mumps at Harvard University and Greater Boston area, which involved 210 confirmed cases, to fit SEIR model parameters. They focused on assessing the impact to two main interventions: 1.) Aggressive quarantine measure based on symptoms alone the absence of a diagnostic test, and 2.) Strict isolation of a suspected cases. To test the robustness of their analysis, they included three models of increasing complexity, finding that the “full intervention” model including the updated diagnosis protocol best fit the data. Most significantly, they find that the updated diagnosis protocol decreased the size of the Harvard

outbreak by approximately three-fourths, which reinforces the notion that preventing cases early on pays dividends later in the epidemics.

Overall, I find their model straightforward, parsimonious, and easy to follow. Further, the inclusion of the two simpler models made the results for the full model more compelling. I feel that the results are of significant interest, especially in the post-COVID era as they can help administrators make hard decisions about stronger mitigation strategies earlier on in an epidemic response. I only have a few general questions and discussion points as well as some minor comments.

We are very glad to hear that reviewer #2 finds our results compelling and of significant interest. We appreciate that our approach is described as parsimonious, we think that's a great qualifier for our models, and we have incorporated it into the main text.

Regarding case definitions (line 188), it states that confirmed cases are those with a positive laboratory test for mumps virus, and probable cases are those who either tested positive for the anti-mumps IgM antibody or had an epidemiologic linkage to another probable or confirmed case. I feel that the "suspected case" definition used in the "updated diagnosis protocol" should be added to this as well, and it should be stated what "clinical symptoms of mumps" were used to identify them. I am also interested to know if false positives from "suspected cases" were ever quantified/estimated through follow-up serological testing. Depending on the disease and the severity of the intervention, the false positive rate should definitely be weighed in the context of false negatives.

We added the clinical definition of suspected case from MDPH (see line L90 in main text), Unfortunately we do not have data on false positives, but added a comment regarding this in the limitations section (L387).

One interesting point to consider is that you assume the parameter q to be generally linear in its effect i.e., the more aggressive you are in case finding and quarantining symptomatic individuals, the more cases you prevent. However, there could be an inflection point where too aggressive of a measure could result in less case finding because of students' decreased self-reporting. I feel this phenomenon was common during the COVID pandemic as individuals were apprehensive about being subjected to isolation and quarantine measures, which translates into financial loss. During the mumps outbreak, were there any anecdotes or evidence that students were apprehensive about reporting their symptoms? Perhaps instead, individuals were self-isolating. I am wondering what impact this would have in your model, although it may be hard to determine the time to "breaking point" when the intervention seemingly backfires. Similarly, was there any indication that students were getting boosters on their own and possibly reducing the pool of susceptibles?

Thanks for making this observation. While we agree that more complex forms for the removal rate modeling such effects could be considered, we think that the high reporting rate makes this

scenario unlikely, at least within the population in our study. We added this consideration explicitly in the discussion (L341-349).

Minor:
Line 114 – errant “the”

Thanks for noting this typo, it has been corrected.

Line 224 – I had a feeling based on experiences managing COVID on a university campus that emails would not be very effective. Is there any empiric data to support this, possibly in the form of a survey?

We don't have empirical data to support this conclusion either, but we agree with it based on anecdotal experience by some of the co-authors (who were students at Harvard during the time of this outbreak). In general, students are overwhelmed with many email announcements from the school, which likely reduces the effectiveness of such campaigns. This seems like a topic of active research, see for example this widely cited paper from some years ago:
<https://www.ncbi.nlm.nih.gov/pmc/articles/PMC4248563/>

Can the MMR vaccine be used as post-exposure prophylaxis??

According to CDC's page on the MMR vaccine
(<https://www.cdc.gov/vaccines/vpd/mmr/public/index.html>):

“Unlike with measles, MMR has not been shown to be effective at preventing mumps or rubella in people already infected with the virus (i.e., post-exposure vaccination is not recommended)”

But:

“If you do not have immunity against measles, mumps, and rubella and are exposed to someone with one of these diseases, talk with your doctor about getting MMR vaccine. It is not harmful to get MMR vaccine after being exposed to measles, mumps, or rubella, and doing so may possibly prevent later disease.”

So, the answer would be yes.

Appendix B

Dear RSOS Editors,

We are pleased that our manuscript has been accepted for publication after addressing the last round of minor comments from the reviewers. Please see our point-by-point responses below.

Best regards,
The Authors

Reviewer 1

This is my third review of this paper. It has been enjoyable to observe its progression and improvement over the past year, and I thank the authors for their edits. I am now satisfied with the edits the others have made to the manuscript—most especially moving the details of the methods to the supplementary materials, the tightening of the manuscript, the addition of the very helpful Fig2, and the thorough address of the effective population size and assumptions related to R_0 and R_e . I have just a few minor comments that can be easily addressed below.

We deeply appreciate the continued review from reviewer #1 since the original version of our manuscript. Her excellent comments and suggestions have been critical for us to arrive at the final version of the paper that will reach the scientific community.

Also, I'll note that the 'response to reviewers' refers to R1 as a 'he' but I am a 'she'. Best not to make assumptions!

We apologize about this oversight. We tried our best to keep the responses gender neutral, but one male pronoun invertedly slipped. Thanks so much for bringing this up, so we are more attentive next time.

Line 86: "confirmed cases are those with a positive laboratory test for mumps virus." – is this always a PCR test? If so, please specify

The co-author who oversaw the HUHS response confirmed that the tests were always PCR. We have clarified this in the text.

Line 111: you say that the second intervention was to "treat and isolate" anyone suspected of mumps but this seems to be more intervention #3. It seems to me that there are three interventions, which can be grouped as (1) awareness raising, (2) case identification and testing, (3) isolation of infectious individuals. I recommend you edit this intro section to reflect this.

This is a much better organized listing of the interventions, so we have used the suggested list in the text. Thanks a lot.

Line 187: It is fine to use a larger population size including undergrads, grad students, etc, but you do specify that the 53% susceptible corresponds to 20-29 year olds. Presumably your total population includes individuals above (staff) and below (undergrads) that age range—is the 53% still valid? Just an acknowledgement of the limits of this assumption would be fine

We think it's a fair assumption to use the 0.53 factor in the calculation of the effective population, but it is definitely an approximation due to the fact pointed out by the reviewer, that is, some members of the population are outside of the age range that factor applies to. It's worth noting that the 0.53 value is in turn an estimation in Grad and Lewnard's work.

Line 247: typo: "a the lower".

This typo was corrected.

Line 271: But isn't this assuming that the population size remains constant? And students would typically leave for the summer, so you'd likely get an end to the local epidemic anyhow, wouldn't you? It should be mentioned here that the projected epidemic without intervention is under assumptions in which the population size is not changed.

Great point! There are two things going on here. First, as the reviewer brings up, a basic assumption in our model is the constant population size. This assumption holds throughout the entire modeling and analysis work, and the way we deal during the vacation terms, which had a substantial fraction of the student population away, is by introducing an effective transmission rate that is much lower than the baseline (as calculated from MLE). Second, upon closer inspection of the dates, the peak of the number of new cases when setting $q = 0$ (no change in testing isolation protocol) is at around day 100, which is in fact the day when the summer starts. Due to a much lower transmission rate during the summer, the case count starts to decrease immediately after day 100 (but that's not enough to prevent a much larger outbreak). We rephrased the text in the manuscript as follows:

Finally, the case count in the full model after removing the intervention keeps increasing during the last weeks of the spring term until reaching a maximum of nearly 25 new cases per day at around day 100 (Figure 3B), which is precisely when the summer vacation starts. The case count steadily decreases after that due to the reduced transmission rate during the summer. We interpret the peak as the effect of continuing with the original diagnosis procedure after day 61, since we set $q = 0$ in the model.

Line 285-287: I suggest changing this to say that “the plot SUGGESTS that if the new diagnosis protocol had been implemented within the first 10 days of the outbreak...” rather than ‘reveals’ as we will never really know what might have happened

Absolutely agreed and corrected.

Line 391: I would remove mention of the “post-COVID-19” world as it does not seem like that is ever going to be a reality at this point

Unfortunate, but true. Corrected as well.

Line 178: “times of the values” should be “times the values”
Line 188: “two-doses mumps” should be “two-dose mumps”
Line 226: “three model” should be “models”

Thanks so much for taking the time to go through the supplementary materials and pointing out these typos, all have been corrected.

Reviewer 2

Thank you for the opportunity to review the revised manuscript “Containing the Spread of Mumps on College Campuses” by Mirai and colleagues. The authors sufficiently addressed my epidemiological and translational questions I posed in my primary review. I also appreciated the discussion generated by the comments from Reviewer 1 regarding the model and the layout of the manuscript. Restructuring of the manuscript and moving the methods to the supplementary material significantly improved the readability. The addition of Figure 2 that details the model was also helpful for easy reference. I have not additional comments at this time

Thanks so much for these closing remarks, we are very glad that reviewer #2 found our last round of changes to have improved readability, including the addition of the new figure.